# Fine Cocoa Fermentation with Selected Lactic Acid Bacteria: Fermentation Performance and Impact on Chocolate Composition and Sensory Properties

**DOI:** 10.3390/foods12020340

**Published:** 2023-01-11

**Authors:** Dea Korcari, Alberto Fanton, Giovanni Ricci, Noemi Sofia Rabitti, Monica Laureati, Johannes Hogenboom, Luisa Pellegrino, Davide Emide, Alberto Barbiroli, Maria Grazia Fortina

**Affiliations:** 1Dipartimento di Scienze per gli Alimenti, la Nutrizione e l’Ambiente, Università degli Studi di Milano, 20133 Milan, Italy; 2Rizek Cocoa S.A.S., San Francisco de Macorís 31000, Dominican Republic

**Keywords:** adjunct cultures, improved fermentation, *Theobroma cacao*, proteolytic processes, flavour profile, fine chocolate

## Abstract

Cocoa fermentation is a central step in chocolate manufacturing. In this research, we performed controlled fermentations of a fine cocoa variety to evaluate the impact of adjunct cultures of selected lactic acid bacteria (LAB) on fermentation parameters, chemical composition, and sensory profile of fine cocoa and chocolate. Improved fermentation processes were carried out at the Centre for the Integral Transformation of Cacao (CETICO) in Dominican Republic. Two strains of LAB, previously isolated from cocoa, and belonging to *Lactiplantibacillus fabifermentans* and *Furfurilactibacillus rossiae* species, were employed. Fermentation parameters, protein, peptide and free amino acid profiles of the fermented cocoa and volatile molecules were determined. Sensory analysis of the derived chocolate was also carried out. The obtained results indicated that the addition of the adjunct cultures influences the proteolytic processes and the free amino acid profile. Finally, the adjunct cultures increased the complexity of the flavour profile of the chocolate as they received a higher score for descriptors commonly used for fine chocolate, such as honey and red fruits. The results obtained showed that the selected strains can be an added value to the development of specific flavours that are desirable at industrial level.

## 1. Introduction

Cocoa (*Theobroma cacao* L.) is the main ingredient in the production of chocolate, one of the most important luxury foods with a rapidly expanding market. To become suitable for chocolate production, cocoa seeds extracted from cocoa pods need to undergo a series of transformations, such as fermentation, drying and roasting. Each of these steps plays an important role in the development of the typical chocolate flavour and in the reduction of undesirable notes, such as bitterness and astringency, even though the fermentation process is regarded as the most important post-harvest factor influencing the flavour potential of cocoa [1]. The spontaneous fermentation is characterised by a consortium of naturally occurring yeasts, lactic acid bacteria (LAB) and acetic acid bacteria (AAB) that causes a series of chemical changes within the cocoa bean that are crucial to the development of the complex, typical chocolate flavour [2].

From a quality standpoint, the cocoa beans are classified into “fine” or “flavour” cocoa and “bulk” cocoa. According to the International Cocoa Agreement [3], fine cocoa can be defined as “cocoa that is recognised for its unique flavour and colour”. Generally, fine cocoa is produced from Criollo, Trinitario, and Nacional cocoa varieties, whereas bulk cocoa is obtained from Forastero varieties. Although fine cocoa makes up a small percentage of the cocoa market, estimated around 12%, the demand for this product has increased rapidly in recent years. Only a restricted number of countries export fine cocoa, and Ecuador, the Dominican Republic, and Peru are the main exporters of this product [4].

In order to be classified as fine cocoa, the product has to meet a number of qualitative criteria and the main factor is the flavour. Specifically, fine cocoa is characterised by flavours, such as fruity, flowery, nutty, herbal or caramel, whereas undesirable properties, such as acidity, bitterness, and off-flavour shell will be low. Obtaining products with a standardised sensory profile can be problematic at industrial level, because cocoa fermentation is still carried out using a spontaneous, in-farm process that generally results in end-products of variable quality [5]. For this reason, the use of specific starters or adjunct cultures has been and is still investigated [6,7,8,9,10] with the purpose of standardising the fermentation process. 

The aim of this work is to evaluate the impact of adjunct cultures of selected lactic acid bacteria (LAB) on fermentation parameters, flavour, and sensory profile of fine cocoa and chocolate. In a previous research [11], we characterised the microbial population of spontaneously fermented Dominican cocoa beans, with the aim of selecting autochthonous LAB species, which are used as adjunct cultures that could improve the quality and flavour of the final chocolate. For these reasons, we focalized the attention on two minority LAB species isolated together with the dominant *Lactiplantibacillus plantarum* and *Limosilactibacillus fermentum*. Specifically, we investigated the potential of two strains belonging to *Lactiplantibacillus fabifermentans* and *Furfurilactibacillus rossiae* species. The two selected strains showed interesting activities that could be exploited in a controlled fermentation process, comprising the potential ability to produce cocoa-specific aroma precursors. For this reason, in this study, we carried out improved fermentation processes using as inoculum *L. fabifermentans* strain SAF13 and *F. rossiae* strain SAF51 in comparison with spontaneous cocoa fermentation. The fermentation processes were carried out at the Centre for the Integral Transformation of Cacao (CETICO) in the Dominican Republic. Sensory analyses of the chocolate from the fermented cocoa were carried out for testing the ability of the selected strains to modify the flavour and sensory profile of a fine Criollo cocoa variety. The obtained results showed the potential of the selected strains to be used for a controlled cocoa fermentation, with an improved safety and quality of the process.

## 2. Materials and Methods

### 2.1. Strain Growth and Maintenance

Two LAB strains, *L. fabifermentans* SAF13 and *F. rossiae* SAF51, previously isolated from spontaneously fermented Dominican cocoa, were cultivated in Man Rogosa Sharpe (MRS) broth (Difco Lab., Augsburg, Germany) at 30 °C for 24 h. The strains were maintained in MRS with 150 g/L glycerol at −80 °C and were deposited at the Culture Collection of the Department of Food, Environmental and Nutritional Sciences, University of Milan, Italy. Four mould strains, previously isolated from a contaminated batch of cocoa at the end of the fermentation period and deposited at the Culture Collection of the Department of Food, Environmental and Nutritional Sciences, University of Milan, Italy, were used for the biocontrol assay. The mould strains were cultured in malt extract agar (MEA) plates (Thermo Fisher Scientific, Waltham, MA, USA). Spores were collected by pouring sterile physiological solution (9 g/L NaCl) on the plates after complete sporification, slowly agitating and storing in sterile tubes.

### 2.2. Fermentation Setup and Control

For the inoculum, LAB cultures, grown as reported above, were centrifuged at 7000× *g* for 20 min at 4 °C and the cellular pellet was resuspended in a 50 g/L sucrose solution to a final cell concentration between 10^8^ and 10^9^ CFU/mL. The vitality of the suspension was evaluated by plate counting in MRS agar incubated at 30 °C for 48 h. The cell suspension obtained was used to inoculate 60 kg of freshly harvested cocoa beans of the Criollo variety, provided by Rizek Cacao S.A.S., San Francisco de Macorìs, Dominican Republic. The final cell concentration for each strain, ranging between 10^6^ and 10^7^ CFU/g, is shown in Table 1. The fermentation was carried out in closed plastic containers that were perforated to allow an adequate pulp drainage. The fermenting mass was mixed every 48 h for a total of 6 days of fermentation, and then it was sun dried to a final moisture content lower than 8%. Two different inoculation setups were performed: in the first setup, the inoculation was carried out at the beginning of the fermentation, whereas, in the second setup, the inoculation was performed both at the beginning and after 48 h. A mixed culture fermentation comprised of both strains was also set up to explore the synergies between the strains. A control fermentation without inoculum was performed for comparison (Table 1). For each trial, total LAB and yeast counts were performed in MRS agar with 1 g/L cycloheximide (Merck, City, Rahway, NJ, USA) and Sabouraud Dextrose Agar with 1 g/L tetracycline (Merck, City, Rahway, NJ, USA), respectively. The temperature was recorded daily. The pH of the pulp was evaluated by resuspending 10 g of cocoa in 90 mL of distilled water. Similarly, the cotyledon pH was measured by resuspending 10 g of pulp-free cocoa beans in 90 mL of distilled water. 

### 2.3. Mould Growth Inhibition

Four mould species, isolated from a contaminated batch of cocoa at the end of the fermentation period and belonging to *Aspergillus tamarii*, *Aspergillus nidulans*, *Rhizomucor pusillus* and *Lichteimia ornata,* were tested. To assess the ability of the inoculated strains to prevent mould growth during fermentation, 20 g cocoa samples were retrieved at the end of the 6-day fermentation period and were inserted in sterile 50 mL tubes. A 0.5 mL suspension of approximately 10^4^ fungal spores was added to each tube, which were incubated at 30 °C for 3 days. A non-inoculated sample was also incubated as a control. The growth of mould was evaluated visually.

### 2.4. Protein Extraction and Electrophoretic Characterisation

The electrophoretic characterisation of cocoa bean’s proteins was adapted from Kumari et al. [12]. The seed coats of beans were removed by peeling them off with a scalpel, and beans were then finely grinded. Protein’s extraction was performed by suspending 250 mg of finely grinded cocoa beans in 1 mL of 0.1 mol/L Tris-HCl pH 8.1, in presence of 10 g/L DTT and 10 g/L SDS. The suspension was heated in boiling water for 10 min and stirred for 90 min at room temperature and then centrifuged at 13,000× *g* for 20 min at 4 °C. The surfaced lipids were removed, and the liquid phase, containing the solubilised proteins, was collected. Electrophoretic protein pattern of cocoa beans was characterised through a NuPAGE^®^ electrophoresis system (Invitrogen by Thermo Fisher Scientific, Monza, Italy) by using NuPAGE^®^ 4–12% Bis-Tris Gel and NuPAGE^®^ MES SDS Running Buffer. Samples were prepared in denaturing buffer (0.125 mol/L Tris-HCl, pH 6.8, 500 mL/L glycerol, 17 g/L SDS, 0.1 g/L bromophenol blue, containing 10 g/L 2-mercaptoethanol (2-ME), and run according to manufacturer’s instructions for reduced condition (Invitrogen by Thermo Fisher Scientific). At the end of the run, proteins were fixed by soaking the gel in 50 g/L glutaraldehyde and 500 mL/L methanol for 60 min, whereupon the gel was stained in Coomassie Blue.

### 2.5. Chromatographic Characterisations

Peptide profiles were studied by chromatographic techniques. Samples were prepared by suspending 250 mg of finely grinded cocoa beans in 5 mL of PBS. Suspensions were stirred for 60 min at 4 °C and then centrifuged at 10,000× *g* for 40 min at 4 °C. The liquid supernatant phase was collected, added of 20 mL/L trifluoroacetic acid (TFA) and centrifuged at 13,000× *g* for 10 min at 4 °C. The clear supernatant was used for the chromatographic characterisations. Size exclusion high performance liquid chromatography (SE-HPLC) separations were performed in a Superdex^TM^ Peptide 10/300 GL column (Cytiva Europe GmbH, Milano, Italy) fitted on a chromatographic apparatus composed by a Waters 600E multisolvent delivery system and a Waters 2487 Dual λ Absorbance Detector (Waters, Sesto San Giovanni, Italy). Mobile phase was PBS, at 0.5 mL/min. Before analysis, the pH of the sample was corrected at around 7 by using 5 mol/L NaOH. The same supernatant was analysed by reverse phase high performance liquid chromatography (RP-HPLC). Separations were performed in a SIMMETRY300 C18 (5 μm) (4.6 × 250 mm) column (Waters) fitted on a chromatographic apparatus composed by two Waters 510 HPLC pumps, a Waters 717plus Autosampler, and a Waters 996 Photodiode Array Detector (Waters). Separations were run at 0.8 mL/min, mixing solution A (1 mL/L TFA in water) and solution B (1 mL/L TFA in acetonitrile) as follows: 5 min isocratic 100% solution A, as well as 50 min linear gradient to 70% solution A and 30% solution B.

### 2.6. Free Amino Acids Profile

For the extraction of free amino acids, 4.5 g of finely grinded dry cocoa beans were dispersed under magnetic stirring in 40 mL of sodium citrate buffer (0.2 mol/L, pH 2.2) for 40 min. The dispersion was sonicated for 5 min and filtered using a Whatman No. 41 filter (Whatman plc, Maidstone, UK). An aliquot of 10 mL of the filtrate was deproteinated by adding 10 mL solution of 75 g/L sulfosalicylic acid at pH 1.75, under magnetic stirring, for 5 min. After the addition of 250 µL of Nor-Leucine as internal standard the solution was brought to the final volume of 25 mL and filtered using a Whatman No. 42 filter (Whatman) and subsequently using a 0.2 µm pore size syringe filter. A 100 µL volume of the filtered sample was used for quantification of free amino acids by ion exchange chromatography (IEC) using a Biochrom 30+ chromatograph equipped with an automatic sampler, as described by Hogenboom et al. [13]. An unfermented dry cocoa sample was also analysed to evaluate the changes in the concentration of free amino acids due to the fermentation process.

### 2.7. Cocoa Liquor and Chocolate Preparation

Cocoa liquor and chocolate samples were prepared at the KahKow Experience, Santo Domingo, Dominican Republic. Dried cocoa beans were roasted at 100 °C for 20 min. Roasted beans were then cracked and dehulled using an automatic winnower. The cocoa nibs were ground to obtain cocoa liquor. The formulation used to produce the chocolate samples was: cocoa liquor 58%, deodorised cocoa butter 10%, and powdered white sugar 32%. After refining through a roller press, the mixture was conched (50 °C, 24 h) and tempered by heating the mass at 50 °C and cooling at 31 °C. Finally, the chocolate was moulded into 7 g pieces that were individually wrapped and stored at 20 °C until the sensory evaluation.

### 2.8. Volatile Molecules Assay 

To analyse the aromatic compounds of the cocoa liquor, 100 mg of sample were inserted in 20 mL headspace vials. Each sample was concentrated in a Gerstel MPS 2 with Dynamic Headspace DHS instrument (GERSTEL GmbH & Co., GERSTEL Inc., Linthicum, MD, USA) at 70 °C under 750 rpm agitation. The volatile fraction was trapped with helium (10 min, 20 °C) and analysed in GC-MS (Agilent GC 6890N, Agilent Technologies Inc., Santa Clara, CA, USA) equipped with an Agilent DB 624 column (60 m × 0.32 mm × 1.80 µm) after thermodesorption (heat rate 30–260 °C). The carrier gas was helium at a flow rate of 2 mL/min. The oven temperature was set at a heat rate of 40–255 °C. The mass detector was an Agilent MSD 5975 with an electronic ionisation at 70 eV. The data were analysed using the Agilent ChemStation software, Gerstel Maestro Version (Linthicum, MD, USA).

### 2.9. Sensory Analysis of Chocolate

In order to describe the chocolate samples’ sensory properties, the sensory profile method was applied [14]. Ten subjects (7 women and 3 men aged between 20 and 33 years) were selected from a panelist pool consisting of University of Milan students and employees. To participate in the study, subjects had to prefer and consume dark chocolate. Written informed consent was obtained from each subject. The protocol was approved by the Ethics Committee of the University of Milan (n. 32/12). The method consisted of a preliminary training phase to acquire familiarity with the product and the methodology, followed by a second phase focused on the sample evaluation. Subjects were trained over a period of one month (nine sessions of approximately 1 h each). During this phase, several commercial dark chocolates (including samples from Criollo variety) were selected and presented to the assessors in order to provide a wide range of sensory variability for each attribute and thus stimulate the generation of descriptors. As training progressed, descriptive terms and relevant reference standards were defined through panel discussion. Fourteen sensory descriptors covering appearance (brown colour), odour (cocoa, red fruits, toasted, honey), taste (sweet, bitter, sour), flavour (cocoa, red fruits, toasted, honey) and mouthfeel sensations (hardness, astringency) were defined (see Appendix A). Once the vocabulary was set up, the assessors performed three preliminary sessions to acquire familiarity with the scale. After the training phase, assessors evaluated the six chocolate samples, named control, A, B, C, D, and E (according to Table 1) in three replicates (of which two were retained for data analysis). The three replicates were performed in different days at the sensory laboratory of the Department of Food, Environmental and Nutritional Sciences (University of Milan) designed according to ISO guidelines [15]. Assessors were asked not to smoke, eat, or drink anything, except water, for 1 hour before the tasting sessions. 

Assessors received one 7-g piece of each sample, for a total of six chocolates in each tasting session and rated the intensity of each sensory attribute using a 9-point scale (“1” = “absence/minimum intensity of the descriptor”; “9” = “highest intensity of the descriptor”). Chocolate samples were served at room temperature in plastic plates that were coded with 3-digit numbers. The order in which the samples were presented was systematically varied over assessors and replicates in order to balance the effects of serving order and carryover [16]. Chocolate samples were evaluated in individual booths under white light. The assessors were provided with unsalted crackers and water to clean their mouths during the evaluation. The data were collected using Fizz software v2.47 (Biosystemes, Couternon, France). 

### 2.10. Statistical Analysis

Experiments were performed in triplicate, and results are shown as mean ± standard deviation. *t*-tests were performed using GraphPad Prism 8 (v. 8.4.3, GraphPad Software San Diego, CA, USA) and significance levels are indicated as n.s. for non-significant differences, one asterisk (*) for *p* < 0.05, two asterisks (**) for *p* < 0.01, or three asterisks (***) for *p* < 0.001. Sensory data were first analysed by means of 3-way ANOVA considering samples (6), judges (10), replicates (2), and their relevant second order interactions as factors and sensory attributes as dependent variables. When the ANOVA showed a significant effect (*p* < 0.05), the least significant difference (LSD) was applied as a multiple comparison test using the statistical software program STATGRAPHICS PLUS version 5.0 (Manugest KS Inc., Rockville, MD, USA). In order to examine the results from a multidimensional point of view, principal component analysis (PCA) was then performed on sensory data averaged across judges and replicates (matrix Samples × Descriptors) using XLSTAT version 2019.2.2 (Addinsoft, Boston, MA, USA). 

## 3. Results

### 3.1. Fermentation Parameters

To study the progress of the fermentation, the temperature and pH of the cocoa samples were recorded daily. Counts of LAB and yeasts were also performed during the initial 72 h of fermentation; after this period, the LAB and yeast counts were <10 CFU/g of cocoa. The results for each fermentation setup are reported in Figure 1. The inoculation of the selected strains did not affect the temperature or pH of the pulp and cotyledon. Whereas the strain *L. fabifermentans* SAF13 did not significantly impact the total LAB or yeast counts, the inoculum of the strain *F. rossiae* SAF51 led to a significant increase in the total LAB count after 24 h of fermentation, but this difference was not observed in the subsequent days. The inoculum of this strain after 48 h of fermentation led to a significant decrease in total yeast count at the 72-h mark. Similarly, the inoculum of the consortia of the two strains did not have a significant impact on the fermentation parameters considered.

The pH of the pulp did not significantly change during the first 24 h of fermentation, despite the organic acid production by the LAB. This aspect is related to the consumption of citric acid by the LAB, which is known to raise the pH of the cocoa pulp at the beginning of the fermentation [17]. The pH was lowest at 48 h, then it increased up to 4–4.5. The pH of the cotyledon, on the other hand, steadily decreased from the initial pH of 5–5.5 to about 4.5, matching the pH of the pulp. The temperature profile depended on the mixing of the fermenting mass, as it rose significantly after 48 h when the first mixing was performed, to reach a temperature of 45–50 °C that was maintained throughout the rest of the fermentation period. 

The LAB appeared to have their maximum growth at 24–48 h from the beginning of the fermentation, whereas the growth of yeasts appeared to decrease earlier compared to LAB, which may be due to the higher sensitivity of yeasts to the stress factors, such as the low pH and high temperature that start to take place after 48 h of fermentation.

### 3.2. Mould Growth Inhibition

The results of the in situ antifungal activity towards mould strains isolated from contaminated cocoa are represented in Figure 2. Both strains showed some inhibition but, while the four mould strains were completely inhibited by *L. fabifermentans* SAF13, no growth was observed after three days of incubation, and the strain *F. rossiae* SAF51 only delayed the growth. On the other hand, all four mould species had a good ability to grow in the spontaneously fermented control.

### 3.3. Protein and Peptide Characterisation

The unfermented sample showed the peculiar protein profile of *Theobroma cacao* seeds. In the protein pattern (Appendix A), it is possible to identify the Vicilin-type storage proteins (corresponding to the bands at 47, 31 and 14.5 kDa) and the albumins (corresponding to the band at 21 kDa) [18]. During the fermentation, most of the proteins are hydrolysed: vicilin-type proteins showed a complete degradation [12], whereas some albumins were still visible, indicating that proteolysis of this protein was only partial [18]. An accumulation of relatively large peptides was observed around 5-7 kDa, together with a light band at 3.5 kDa. No smaller peptides were observed in the gel (Appendix A).

Low molecular weight peptides (<5 kDa) have been extracted from fermented and unfermented cocoa beans and characterised by chromatographic approaches. SE-HPLC showed a modest accumulation of peptides at dimensions just higher than 500 Da (Appendix A). According to their molecular weights, these peptides could correspond to the band visible at 3.5 kDa in the electrophoretic separation.

A more efficient separation of peptides was achieved by RP-HPLC. Although this technique does not give information about peptide sizes, the chromatograms showed the presence of three main peaks in the unfermented sample, eluting at 21.5 (peak 1), 23 (peak 2), and 24 (peak 3) minutes, respectively (Figure 3a and Appendix A), all having an absorption spectrum with a maximum at 280 nm. The area of these peaks decreased in the fermented samples, suggesting that they refer to three peptides already present in the unfermented sample, digested by proteases during fermentation. Interestingly, the area of these peaks can be correlated to the inoculum and fermentation protocol applied. The largest area reduction was observed with the spontaneous fermentation (Figure 3b), while, in the presence of the double inoculum (SAF13 + SAF51 at t_0_ and t_48_), the reduction in area was close to that observed with spontaneous fermentation. Noteworthy, the inoculum of *L. fabifermentans* SAF13 or *F. rossiae* SAF51 was observed separately, resulting in an intermediated reduction of the areas, with the inoculum at t_0_ + t_48_ more active than the t_0_ alone.

### 3.4. Free Amino Acid Profile

The chromatographic profiles of free amino acids in the unfermented cocoa and the spontaneously fermented control are shown in Figure 4. The obtained data, including those of A–E samples after six days of fermentation, are compiled in Table 2, expressed as mg/g of total protein. The total amount of free amino acids in the fermented samples was almost twice that of the unfermented sample. The inoculation of the strain *F. rossiae* SAF51 appears to promote the liberation of amino acids when compared to the samples inoculated with the strain *L. fabifermentans* SAF13. The fermentation protocol influenced the final concentration of single amino acids and the total amount of amino acids slightly decreased in samples inoculated at both t_0_ and at t_48_. The amount of some amino acids decreased during the fermentation (Asn, Gln, Gaba, His, Arg), some remained unchanged (Asp, Glu, Pro), and other increased significantly (Thr, Met, Leu Phe, Lys). 

The amino acid profile of the samples inoculated with the strain *L. fabifermentans* SAF13 was similar to the one inoculated with the strain *F. rossiae* SAF51 but, when considering the relative proportion of the single amino acids, the samples inoculated with the strain SAF13 had a consistent abundance of Asn, Glu, and Ala, whereas the samples inoculated with SAF51 contained a higher relative amount of Ile, Tyr, Orn, and Lys. The sample inoculated with the consortia *L. fabifermentans* SAF13 and *F. rossiae* SAF51 had a profile that closely resembled that obtained with *F. rossiae* SAF51 alone.

### 3.5. Volatile Molecules of Cocoa Bulk

The results obtained are reported in Figure 5. The flavour descriptors for each molecule are reported according to The Good Scents Company, which is dedicated to providing information for the flavor, food and fragrance industry (http://www.thegoodscentscompany.com/, accessed on 10 February 2022). The amount of each volatile molecule detected is reported in Appendix A. In general, the samples inoculated with the strain *L. fabifermentans* SAF13 presented a higher total amount of volatiles when compared to the control, whereas the amount was lower in samples inoculated with the strain *F. rossiae* SAF51. In the samples inoculated with *L. fabifermentans* SAF13, the compounds that increased the most, with concentrations more than twice that of the control, were 2,3-butanedion, hexanal, 2-heptanone, and 2-nonanone. Furthermore, in the sample inoculated at t_0_ only, an increase of approximately 50% was recorded for 2-methyl butyric acid and 3-methyl butyric acid. On the other hand, a reduction was recorded for benzyl acetaldehyde, tetramethyl pyrazine, and phenethyl acetate. 

Considering the samples inoculated with the strain *F. rossiae* SAF51, a reduction in the majority of compounds was noted, which was more pronounced in the sample inoculated both at t_0_ and at t_48_. Only 2-methyl butyric acid and 3-methyl butyric acid increased in both samples. The sample inoculated with both strains shows an increase in the concentration of hexanal and 2-nonanone, probably due to the high production of these compounds in presence of *L. fabifermentans* SAF13, as previously mentioned. Significant differences were also observed for the acetic acid amount in the different samples. The fermentations performed with *F. rossiae* SAF51, as well as the one performed by the consortia of the two strains, produced lower acetic acid amounts when compared to the control; the samples inoculated with the strain *L. fabifermentans* SAF13, on the other hand, contained higher amounts of acetic acid. 

### 3.6. Sensory Analysis

Three-way ANOVA results indicated that the panel of assessors was reliable (Appendix A) and that all the sensory descriptors significantly (*p* < 0.05) discriminated the samples. Since ANOVA results indicated that the mean scores for each sample given by the panel for each attribute could be assumed satisfactory estimates of the sensory profile of samples (Appendix A), sensory data were averaged across assessors and replicates and submitted to PCA. According to correlation loadings plot, variables with less than 50% explained variance were left out from the analysis (i.e., red fruits odour, astringency, and bitter). The biplot based on samples and the remaining variables is shown in Figure 6. The variance explained by the first two principal components was 80.80%. Moving from left to right along the first component (explained variance 64.56%) of Figure 6, the control sample and samples A and E were separated from the rest of the samples. The second component (explained variance 16.24%) distinguished samples C from E. The control sample (positioned in the upper left pane, Figure 6) was mainly described by brown colour, cocoa odour and flavour, sourness, and hardness and only partially by roasted odour and flavour. Samples A and E were mainly perceived with high intensity of roasted odour and flavour, as well as red fruits flavour. These samples (A, E, control) had also the lowest intensity of sensory properties (honey odour and flavour and sweetness) located in the positive part of PC1 of Figure 6, which mainly characterise samples D and B. Sample C (located in the upper right pane, Figure 6) was mainly sour.

## 4. Discussion

Controlled cocoa fermentations have received a strong interest from chocolate manufacturers and cocoa producers as a natural way to obtain specific flavour profiles without the need of using additives or flavourings. Although a lot of effort from the scientific community has been put into the study of different yeast species, the research is still lacking when it comes to LAB, especially in the study of minor species that do not dominate the fermentation but that may play a role in the flavour profile of the final product. Furthermore, there is a gap in knowledge about the optimal fermentation parameters to apply in a controlled fermentation and their role in the quality of the final product.

In this research, we performed controlled fermentations of a fine flavour cocoa variety using two selected LAB strains, *F. rossiae* SAF51 and *L. fabifermentans* SAF13, as adjunct cultures. Two different inoculation setups were performed: the fermenting cocoa was inoculated either at t_0_ only, or both at t_0_ and at t_48_; the second inoculation was performed because, from the available scientific literature, LAB are indicated to develop and perform at their optimum after the first stage dominated by yeasts [1]. The obtained results show that, whereas the two strains did not significantly impact fermentation parameters such as the temperature and the pH, they gave different profiles when considering the final product. The fact that the fermentation parameters were similar to those of a typical well-fermented cocoa is a positive attribute, as they are crucial for the outcome of the fermentation process. Furthermore, there was no excessive acidity that is perceived as the main drawback of the presence of LAB in cocoa fermentations [19]. Indeed, the sensory profile showed that the spontaneous control was the sample with the highest perceived acidity (Appendix A). This can be partially due to the heterofermentative phenotype of the LAB used, which produce different metabolites depending on the carbon source used. Furthermore, the GC–MS analysis of the cocoa liquor showed a different concentration of acetic acid in the fermentation with each strain, so an interaction with the microbiota as a whole can also be hypothesised, given that acetic acid bacteria are the main producers of acetic acid.

Free amino acids, di- and tripeptides, are some of the most important precursors of flavour formation in chocolate. Not only they participate in the Maillard reaction during roasting, giving the typical chocolate flavour, but they also go through degradation to produce aldehydes, ketones, and other volatile molecules that have a significant impact in the flavour profile [20]. Vicilin-type proteins were found to be selectively degraded by endogenous proteases under fermentation conditions where optimal levels of aroma precursors were obtained [21]. Our results showed that cocoa proteins are digested almost completely whether the fermentation occurs spontaneously or is driven by the inoculum of *F. rossiae* SAF51 and/or *L. fabifermentans* SAF13. On the other hand, the inoculum influences the extent and the kinetics of the proteolytic processes occurring on some protein fractions: competition seems to take place between inoculated LAB and spontaneous microflora, since the digestion of some peptides has apparently been reduced compared to the spontaneous fermentation. These results are consistent with our recent evidence [11] that *F. rossiae* SAF51 and *L. fabifermentans* SAF13 strains have a different peptidasic and proteolytic set, even compared to *L. plantarum*, the predominant LAB in cocoa fermentation. The addition of the adjunct cultures affected the free amino acid profile, either due to the peptidases from the LAB that may be released after the cell death or from the creation of conditions closer to the optimum for the endogenous aspartic proteases and carboxypeptidases that have been suggested as the main enzymes that cleave and release amino acids in cocoa. Hydrophobic amino acids, such as alanine, tyrosine, and phenylalanine, have been indicated as the main precursors of cocoa flavour components [22,23]. Their concentration increased with cocoa fermentation, and the inoculation of the strain *F. rossiae* SAF51 appeared to promote their accumulation. The amino acid liberation kinetic observed during cocoa fermentation has been previously reported by Kirchhoff et al. [24], which has also noted the elevated proportion between hydrophobic and acidic amino acids [25] that, in our samples, is approximately 70:15. More recently, Deus et al. [26], monitoring on-farm cocoa fermentation at 12-h intervals, reported the total free amino acid content to have doubled after six days, comparable to our findings. Interestingly, the authors observed higher amino acid contents after 60 h and 96 h than at the end of fermentation. Similar to our results, hydrophobic amino acids increased to a greater extent than acidic amino acids, resulting in a final proportion between hydrophobic and acidic amino acids of 51.0:12.5. While these authors observed a decrease in glutamic acid after fermentation, our data show this amino acid to remain constant in the control sample and to slightly increase with the inoculum of the selected strains, demonstrating the possibility to direct the fermentation process through selection of the microbial population.

Because the cocoa liquor undergoes significant changes during chocolate manufacturing, especially during conching, no direct correlation can be drawn between the volatile compounds detected in cocoa liquor and the flavour profile of the final product. Despite this, the analysis of the volatile compounds is a useful method to highlight the potential activity of selected strains in a controlled fermentation. The adjunct cultures seemed to increase the complexity of the flavour profile of the chocolate as they received a higher score for descriptors that commonly characterise fine chocolate, such as honey and red fruits (Appendix A). The fermentation protocol also played an important role in all aspects considered, so it is important to take this factor into consideration for future starter culture development. Finally, the cultures had a protective role against mould contamination, especially the strains *L. fabifermentans* SAF13 that inhibited the growth of the tested moulds completely. These data are of interest, as this strain can be a good biocontrol agent of aflatoxin-producing mould species during cocoa fermentation. The possibility of using selected LAB to prevent mould contamination in cocoa has been previously investigated with good outcomes [27,28,29], so this aspect is an added value to the use of LAB in starter cultures for cocoa fermentation.

## 5. Conclusions

In conclusion, this work showed that the selected LAB strains play a significant role in the achievement of the desired flavour and quality of chocolate. Interestingly, significant differences were observed both between strains and between inoculation timepoints. Although their presence may not be essential to obtain a product with an acceptable taste, selected strains can improve the development of specific flavours that are desirable at an industrial level. 

## Figures and Tables

**Figure 1 foods-12-00340-f001:**
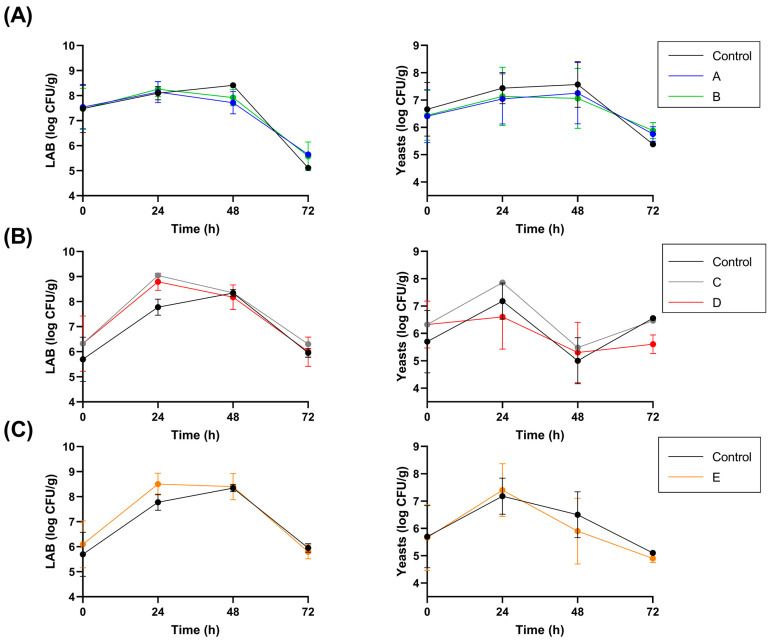
LAB and yeast counts for each inoculated species and different inoculation protocol, in comparison with spontaneous fermentation (black): (**A**) *L. fabifermentans* at t_0_ (blue) and t_0_ + t_48_ (green), (**B**) *F. rossiae* at t_0_ (grey) and t_0_ + t_48_ (red), (**C**) co-culture at t_0_ + t_48_ (orange).

**Figure 2 foods-12-00340-f002:**
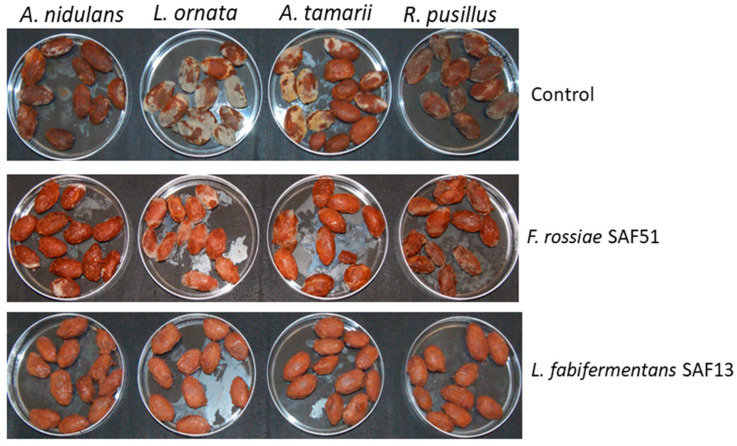
Antifungal activity of strains *L. fabifermentans* SAF13 and *F. rossiae* SAF51 against four mould strains isolated from cocoa. Spontaneously fermented cocoa was used as control. Cocoa samples were retrieved at the end of the six-day fermentation period and incubated at 30 °C for three days after inoculum with fungal spores.

**Figure 3 foods-12-00340-f003:**
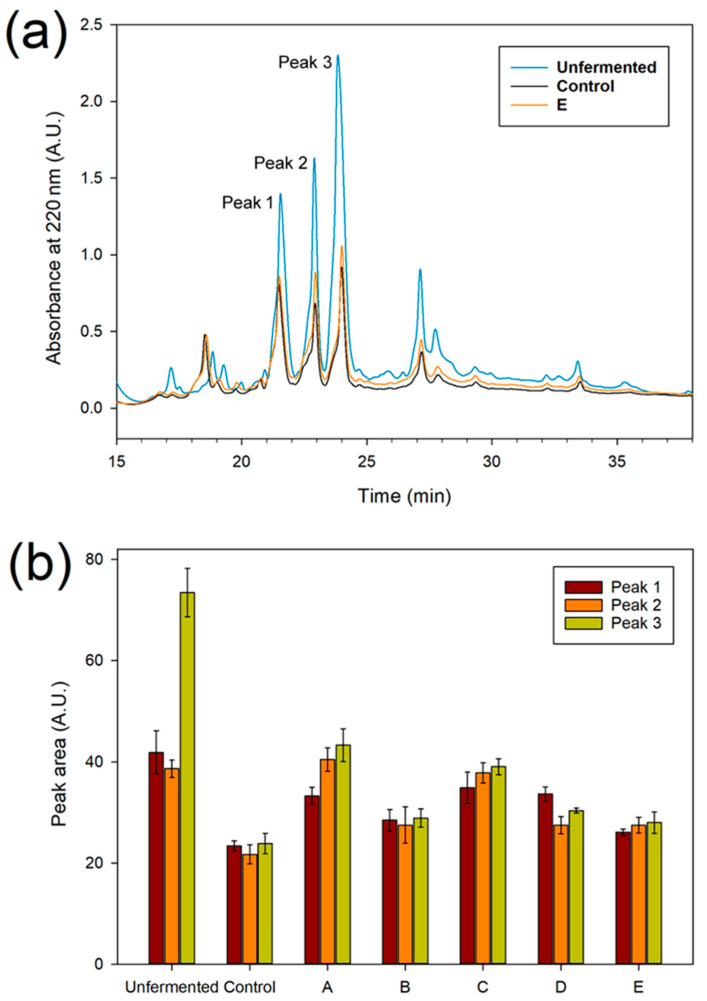
Peptide profile of unfermented and fermented samples. (**a**): RP-HPLC chromatograms of peptide extracted from cocoa beans. For the sake of comparison, only chromatograms from unfermented, control and co-culture at t_0_ + t_48_ have been reported. Chromatograms of all the samples are available in Appendix A. (**b**): Areas of the three main peaks observed in RP-HPLC chromatograms; unfermented: unfermented cocoa beans; control: spontaneous fermentation; A: inoculated with *L. fabifermentans* at t_0_; B: inoculated with *L. fabifermentans* at t_0_ and t_48_; C: inoculated with *F. rossiae* at t_0_; D: inoculated with *F. rossiae* at t_0_ and t_48_; E: inoculated with both strains at t_0_ and t_48_.

**Figure 4 foods-12-00340-f004:**
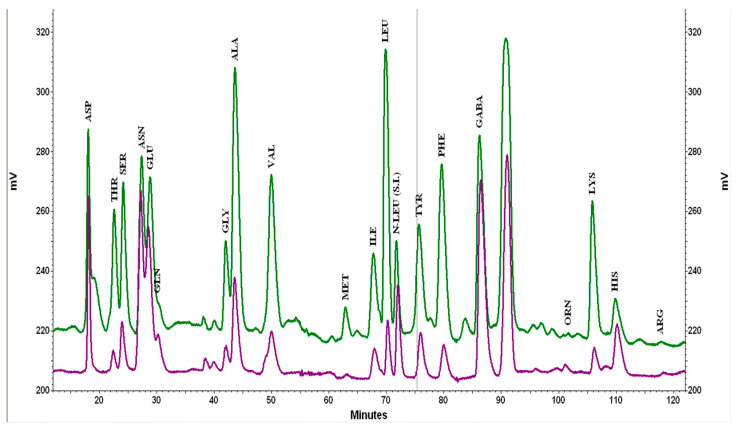
Chromatographic profiles (detection at 570 nm) of the unfermented cocoa (purple) and fermented (six days) control (green).

**Figure 5 foods-12-00340-f005:**
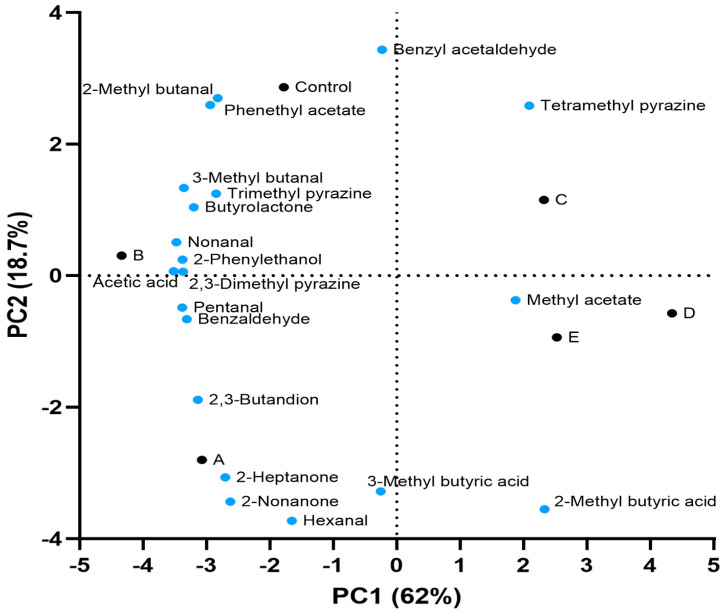
Volatile compounds analysis of cocoa liquor. Control: spontaneous fermentation; A: inoculated with *L. fabifermentans* at t_0_; B: inoculated with *L. fabifermentans* at t_0_ and t_48_; C: inoculated with *F. rossiae* at t_0_; D: inoculated with *F. rossiae* at t_0_ and t_48_; E: inoculated with both strains at t_0_ and t_48_.

**Figure 6 foods-12-00340-f006:**
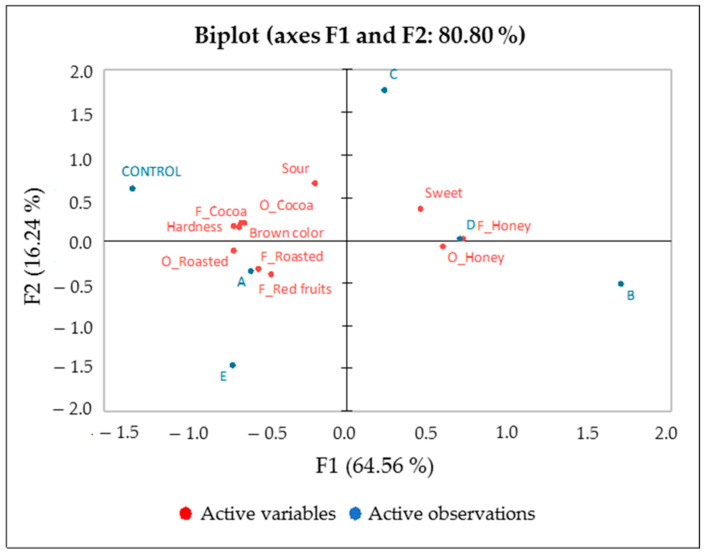
Biplot obtained by the PCA model of chocolate sensory data. A: inoculated with *L. fabifermentans* at t_0_; B: inoculated with *L. fabifermentans* at t_0_ and t_48_; C: inoculated with *F. rossiae* at t_0_; D: inoculated with *F. rossiae* at t_0_ and t_48_; E: inoculated with both strains at t_0_ and t_48_. O indicates odour, F indicates flavour.

**Table 1 foods-12-00340-t001:** Inoculation protocols and microbial counts for each inoculum.

Sample	Fermentation Protocol	Inoculum (CFU/g of Cocoa)
Control	Spontaneous fermentation	-
A	Inoculum with strain *L. fabifermentans* SAF13 at t_0_	6.25 × 10^6^
B	Inoculum with strain *L. fabifermentans* SAF13 at t_0_ and t_48_	6.25 × 10^6^ at t_0_ 1.6 × 10^7^ at t_48_
C	Inoculum with strain *F. rossiae* SAF51 at t_0_	1.4 × 10^7^
D	Inoculum with strain *F. rossiae* SAF51 at t_0_ and t_48_	1.4 × 10^7^ at t_0_ 1.9 × 10^7^ at t_48_
E	Inoculum with strains *L. fabifermentans* SAF13 and *F. rossiae* SAF51 at t_0_ and t_48_	SAF13: 6.25 × 10^6^ at t_0_ +1.6 × 10^7^ at t_48_ SAF51: 1.4 × 10^7^ at t_0_ + 1.9 × 10^7^ at t_48_

**Table 2 foods-12-00340-t002:** Free amino acids in fermented (six days) cocoa beans in comparison with unfermented cocoa and spontaneously fermented cocoa (control), expressed as mg/g of the total protein content.

Sample	Unfermented	Control	A	B	C	D	E
Asp	2.21 ± 0.02 ^a^	3.13 ± 0.03 ^b^	3.20 ± 0.03 ^b^	3.30 ± 0.03 ^bc^	3.70 ± 0.04 ^cd^	3.54 ± 0.04 ^bcd^	3.94 ± 0.04 ^d^
Thr	0.37 ± 0.01 ^a^	2.29 ± 0.03 ^b^	2.21 ± 0.02 ^b^	2.12 ± 0.02 ^b^	2.51 ± 0.03 ^bc^	2.41 ± 0.03 ^bc^	2.74 ± 0.03 ^c^
Ser	0.78 ± 0.01 ^a^	2.49 ± 0.03 ^bc^	2.35 ± 0.03 ^b^	2.27 ± 0.03 ^b^	2.82 ± 0.04 ^cd^	2.50 ± 0.03 ^bc^	3.02 ± 0.04 ^d^
Asn	4.14 ± 0.06 ^abc^	3.74 ± 0.05 ^a^	4.49 ± 0.06 ^c^	4.32 ± 0.06 ^bc^	3.72 ± 0.05 ^a^	3.96 ± 0.06 ^ab^	3.92 ± 0.05 ^ab^
Glu	4.68 ± 0.05 ^a^	4.91 ± 0.06 ^ab^	5.40 ± 0.06 ^c^	5.52 ± 0.06 ^cd^	5.24 ± 0.06 ^bc^	5.52 ± 0.06 ^cd^	5.85 ± 0.07 ^d^
Gln	0.98 ± 0.07 ^b^	0.58 ± 0.04 ^ab^	0.51 ± 0.04 ^a^	0.38 ± 0.03 ^a^	0.48 ± 0.04 ^a^	0.41 ± 0.03 ^a^	0.43 ± 0.03 ^a^
Gly	0.32 ± 0.01 ^a^	1.07 ± 0.01 ^bc^	1.06 ± 0.01 ^bc^	0.92 ± 0.01 ^b^	1.41 ± 0.02 ^c^	1.20 ± 0.02 ^bc^	1.33 ± 0.02 ^bc^
Ala	1.46 ± 0.02 ^a^	4.43 ± 0.07 ^b^	4.62 ± 0.07 ^bc^	4.80 ± 0.07 ^b^	4.95 ± 0.07 ^c^	4.89 ± 0.07 ^c^	4.81 ± 0.07 ^b^
Val	1.24 ± 0.01 ^a^	4.05 ± 0.04 ^cd^	3.74 ± 0.04 ^bc^	3.53 ± 0.04 ^b^	4.68 ± 0.05 ^ef^	4.31 ± 0.05 ^de^	5.03 ± 0.06 ^f^
Met	0.06 ± 0.01 ^ab^	0.79 ± 0.02 ^de^	0.49 ± 0.01 ^bcd^	0.28 ± 0.01 ^abc^	1.22 ± 0.03 ^e^	0.7 ± 0.02 ^cd^	1.07 ± 0.03 ^de^
Ile	0.97 ± 0.02 ^a^	2.42 ± 0.05 ^bc^	2.17 ± 0.05 ^b^	2.07 ± 0.04 ^b^	2.92 ± 0.06 ^de^	2.65 ± 0.06 ^cd^	3.24 ± 0.07 ^e^
Leu	1.30 ± 0.03 ^a^	7.91 ± 0.17 ^bc^	7.49 ± 0.16 ^b^	7.80 ± 0,17 ^bc^	8.52 ± 0.19 ^de^	8.17 ± 0.18 ^cd^	8.74 ± 0.19 ^e^
Tyr	1.77 ± 0.11 ^a^	4.35 ± 0.26 ^bc^	3.97 ± 0.24 ^b^	4.06 ± 0.24 ^b^	5.30 ± 0.32 ^d^	4.63 ± 0.28 ^c^	5.66 ± 0.34 ^d^
Phe	1.28 ± 0.02 ^a^	6.78 ± 0.09 ^c^	6.29 ± 0.08 ^b^	6.21 ± 0.08 ^b^	7.93 ± 0.11 ^d^	6.77 ± 0.09 ^c^	8.12 ± 0.11 ^d^
Gaba	4.77 ± 0.33 ^d^	4.45 ± 0.31 ^cd^	3.74 ± 0.26 ^a^	3.95 ± 0.27 ^ab^	4.63 ± 0.32 ^cd^	4.27 ± 0.30 ^bc^	4.74 ± 0.33 ^d^
Orn	0.21 ± 0.01 ^a^	0.12 ± 0.01 ^a^	0.17 ± 0.01 ^a^	0.12 ± 0.01 ^a^	0.27 ± 0.01 ^a^	0.30 ± 0.01 ^a^	0.27 ± 0.01 ^a^
Lys	0.76 ± 0.01 ^a^	4.40 ± 0.04 ^c^	3.93 ± 0.04 ^b^	3.57 ± 0.03 ^b^	5.32 ± 0.05 ^e^	4.74 ± 0.05 ^cd^	5.16 ± 0.05 ^de^
His	2.57 ± 0.04 ^c^	2.23 ± 0.04 ^abc^	1.82 ± 0.03 ^a^	1.82 ± 0.03 ^a^	2.32 ± 0.04 ^bc^	1.99 ± 0.03 ^ab^	2.41 ± 0.04 ^bc^
Arg	0.29 ± 0.01 ^a^	0.26 ± 0.01 ^a^	0.20 ± 0.01 ^a^	0.14 ± 0.01 ^a^	0.13 ± 0.01 ^a^	0.15 ± 0.01 ^a^	0.19 ± 0.01 ^a^
Pro	1.34 ± 0.03 ^a^	2.19 ± 0.04 ^bc^	2.15 ± 0.04 ^bc^	1.80 ± 0.04 ^b^	2.40 ± 0.05 ^cd^	2.04 ± 0.04 ^bc^	2.82 ± 0.06 ^d^
Total	31.50 ± 0.27 ^a^	62.59 ± 0.53 ^d^	60.00 ± 0.51 ^c^	58.98 ± 0.50 ^b^	70.47 ± 0.60 ^f^	65.15 ± 0.55 ^e^	73.49 ± 0.62 ^g^

A: inoculated with *L. fabifermentans* at t_0_; B: inoculated with *L. fabifermentans* at t_0_ and t_48_; C: inoculated with *F. rossiae* at t_0_; D: inoculated with *F. rossiae* at t_0_ and t_48_; E: inoculated with both strains at t_0_ and t_48_. Different letters in each row indicate statistical differences (Tukey’s test, *p* < 0.05).

## Data Availability

Data are not available in public datasets. Please contact the authors.

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
