# Peer review of "Fine Cocoa Fermentation with Selected Lactic Acid Bacteria: Fermentation Performance and Impact on Chocolate Composition and Sensory Properties"

_foods, 2023, doi:10.3390/foods12020340_

Round 1

Reviewer 1 Report

Review of manuscript no. foods-2043191

Title: Fine cocoa fermentation with selected Lactic Acid Bacteria: fer-2 mentation performance and impact on chocolate composition 3 and sensory properties

Authors: Korcari et al, M.G. Fortina

In the research paper submitted by Korcari et al, the authors describe experimental work on the inoculation of cocoa bean fermentation with two selected lactobacilli, identified in their earlier and recently published study. The authors used an entire repertoire of analytical methods (aside of fermentation product, polyphenol, and lipid analyses…) to be connected to sensory analysis with a panelist group of students and employees of the University of Milan. The authors found out that the addition of the two lactobacilli (LAB) had no major impact on pH and peptide formation (protein degradation), had a moderate (although not very strong) effect on the composition of the volatile substances as measured by GC-MS. They also concluded that the effects varied in dependence of the used LAB and in dependence of whether the adjunct strains were added once (0 h) or twice (0 and 48 h). Ultimately, they got the most pronounced although still mild impact when they added both LAB twice and together. The fermentation experiment and the addition of the used adjunct LABs were technically sound and well described.

A major unclear question is: why did the authors choose those two minority autochtonous strains in the first place? The reader is referred without comments to their previous publication – but the reasoning for choosing those two strains should be clearly pointed out in the current manuscript. Why choose two LAB strains which occur little in a previous fermentation (one of which had never been described in cocoa fermentation at all)? This needs to be clear so that the reader can appreciate the work more. After consulting the previous paper, my understanding is that these two strains did not inhibit yeasts and AAB, consumed alternative main sugars, and showed in vitro inhibition of four co-isolated mold strains (mainly Aspergillus and Mucor). The authors should rather have selected in addition at least on “negative” control strain, i.e. a majority LAB which competed with yeasts for the main substrate sugars. This would have enriched the study.

What was stated for the fermentation experiment cannot be repeated for the assessment of the described analytical methods. This might be particularly due to the fact that this reviewer cannot recognize when samples were taken. On one site, it is stated that the fermentation lasted for six days, but on the other site only data for 72 h maximum were shown in figures. The very elaborative supplementary tables are generally not easy to read. The protein analysis is restricted to one SDS-PAGE with unfermented and (fully?) fermented samples. No differentiation is possible between the different treatment although one might have assumed that at least sample E (all the possible LAB power) showed some difference. Possibly, the differences were visible in intermediate samples? Not shown in the gel. The peptide analysis using reverse phase HPLC is not very informative. Yes, peptides are generally formed during fermentation – but again – no significant difference become evident here. Possibly, because not the right time point was chosen?

The test for the prevention of molding of cocoa beans (Fig. 2) reads a bit like a stand-alone story. Although the photographic representation of molding looks impressive, I do not think that this can be accepted as a clear evidence because there is not quantitative trait included. Aside, it is not clear to me when the images were taken? After one or three or six days? The reader has to go back to the materials and methods section. Figures should be self-explanatory along with their legends. Finally, if they are rather minority, why should they matter for cocoa bean protection from molds?

In general, the explanation of the analytical methods shows a lot of room for improvement. Besides, I have a few specific points to raise:

1.       Title: please use ‘fine-flavour cocoa fermentation’ instead of ‘fine cocoa…’

 2.       Page 1, lines 14-17: this sentence is too long – please divide into two.

 3.       Page 1, lines 20: You suggest that the kinetics of proteolytic processes were studied – this is a major over-estimation of the actual data derived from the actual experiments. Please modify.

 4.       Safety (page 2, line 68): this is restricted to the four molds, right? A great word not really applicable here. What about Salmonella or spore formers?

 5.       Page 2, line 90: The concentration of added LAB strains of 106 and 107 CFU/g of cocoa beans seem astronomically high. This is a dense solution. How could the effects be so relatively minimal? Were other inoculation concentrations used beforehand? And most importantly, were the added bacteria recovered by re-isolation after 24 and 48 h? I might have overlooked this – but this information might help a lot in interpretation of the data.

 6.       Page 3, line 121: please type 13,000 g.

 7.       Figure 1: why were only three days shown here. What happened until day 6 when the beans were actually dried and processed?

 8.       Page 8, line 298: the statement “the reduction was close, but not equal” is unclear. Close to what? Not equal to what?

 9.       Figure 3A: How do we know that those three major peaks represent proteins or peptides? The method is very unclear or not well explained. The only take-home message I can see is: Control and E are equal (and close), meaning that E is dispensable? Then, why publishing this result?

 10.   Page 9, line 310: chromatographic profiles of what?

 11.   Figure 4 and Table 2: Data for which time points are presented here?

 12.   Page 11, line 337: please give source of this “The Good Scents Company”.

 13.   Figure 5: It is not clear which time point these data are representing. Post-fermentation?

 14.   Figure 6: I cannot find the position of the control fermentation. Please add to make clear what has been changed.

 15.   Page 13, lines 407-408: The authors conclude that the used LAB strains did not develop increase acidity although it is generally thought that LAB would do so. Any hints or thought on why this would happen to those two LAB strains? Lack of actual metabolic activity? Disappearance of the individual strains after having pushed back the native LAB strains? Please discuss options in more detail.

 16.   Page 14, lines 459-461: In order to conclude this, the authors need quantitative data aside of the visual inspection of infected beans (with the used four mold strains). Without quantitative data, this conclusion is not justified. An alternative would be to leave out anti-mold findings – as they are anyway not quantitative.

 17.   Supplementary Figure S1A: I am not sure but which band represents the 21-kDa albumin, being the most abundant storage protein in cocoa seeds? Is it possible that the band remaining at around 20 kDa is actually albumin? Please check and modify your texts accordingly.

Reviewer 2 Report

This study has introduced to us the application feasibility of selected lactic acid bacteria for the fermentation of high-quality cocoa. It is an interesting study and the below points still need to be addressed.  

 1.    It is suggested to add the control experiment of unfermented cocoa bean protein extraction and electrophoresis characterization to explore protein changes between fermented and unfermented samples in 2.4.

2.    It is recommended to add a comparison with unfermented samples in 2.5 Chromatographic characteristics.

3.    In Figure 2, there is no fermentation time label, so it is difficult to distinguish the changes in the fermentation process of samples in the picture.

Round 2

Reviewer 1 Report

Thank you for submitting your revised version and for reacting to the reviewer's comments. Quality is nicely improved.